# Depressed Mood as a Significant Risk Factor for Gynecological Cancer Aggravation

**DOI:** 10.3390/ijerph20196874

**Published:** 2023-10-02

**Authors:** Seon-Mi Lee, Jae-Yun Song, Aeran Seol, Sanghoon Lee, Hyun-Woong Cho, Kyung-Jin Min, Jin-Hwa Hong, Jae-Kwan Lee, Nak-Woo Lee

**Affiliations:** 1Department of Obstetrics and Gynecology, Korea University College of Medicine, Korea University Anam Hospital, Seoul 02841, Republic of Korea; tjsal4142@naver.com (S.-M.L.);; 2Department of Obstetrics and Gynecology, Korea University College of Medicine, Korea University Guro Hospital, Seoul 08308, Republic of Korea; limpcho82@gmail.com (H.-W.C.);; 3Department of Obstetrics and Gynecology, Korea University College of Medicine, Korea University Ansan Hospital, Ansan 15355, Republic of Korea

**Keywords:** gynecological cancer, depressed mood, depression, risk factors, cancer aggravation

## Abstract

Background: The aim of this study was to evaluate the relationship between depressed mood and gynecological cancer outcomes, identifying risk factors for cancer aggravation. Methods: This study was a retrospective analysis of gynecological cancer patients (January 2020–August 2022) at Korea University Anam Hospital using Patient Health Questionnaire-9 (PHQ-9). Patients were classified into non-depressed mood (NDM)- and depressed mood (DM)-based scores. Statistical analysis was performed using Student’s *t*-test, chi-square test, Fisher’s exact test, Kaplan–Meier analysis, and Cox regression analyzing using SPSS. Results: Of the 217 participants, the NDM group comprised 129 patients, and the DM group comprised 88. The two-year disease-free survival (DFS) rates showed significant differences (NDM, 93.6%; DM 86.4%; *p* = 0.006), but overall survival (OS) did not (*p* = 0.128). Patients with stage 3 or higher cancer, undergoing five or more chemotherapies, experiencing post-chemotherapy side effects, and depressed mood had an increased risk of cancer aggravation. Conclusions: Appropriate treatment of depressed mood, as well as adequate treatment for advanced gynecological cancer patients, those with numerous CTx., and those with post-CTx. side effects, may contribute to reducing the risk of cancer aggravation.

## 1. Introduction

According to data reported by the 2020 GLOBOCAN, approximately 19.3 million individuals were newly diagnosed with cancer worldwide, and approximately 10.0 million died from cancer [1]. Patients diagnosed with gynecological cancers such as cervical cancer, endometrial cancer, and ovarian cancer and the number of deaths are as follows: the number of patients diagnosed with cervical cancer is 604,127 (3.1%), the number of deaths from cervical cancer is 341,831 (3.4%), the number of patients diagnosed with endometrial cancer is 417,367 (2.2%), the number of deaths from endometrial cancer is 97,310 (1.0%), the number of patients diagnosed with ovarian cancer is 313,959 (1.7%), and the number of deaths from ovarian cancer is 207,252 (2.1%) [1].

Patients diagnosed with cancer experience various emotions, which can be divided into two categories of mood disorders: depressive episodes and manic or hypermanic episodes. Depressive episodes include feelings of sadness, hopelessness, helplessness, and guilt; suicidal thoughts; fatigue; appetite changes; and loss of will to live. Meanwhile, manic or hypermanic episodes involve feeling emotions such as extreme energy and heightened excitement [2,3]. In 1997, the National Comprehensive Cancer Network (NCCN) defined distress in cancer as a multifactorial unpleasant experience of psychological, social, spiritual, and/or physical natures that may interfere with one’s ability to cope effectively with cancer, its physical symptoms, and its treatment [4]. The degree of distress in cancer patients is scored using an NCCN distress thermometer (DT). The DT consists of a total of 11 points ranging from 0 (no distress) to 10 (extreme distress). DT scores ≥4 are considered moderate to severe distress and are actively recommended to receive appropriate management for distress [5]. According to recent statistics reported by the National Cancer Institute, approximately 25% of cancer patients suffer from depressive episodes, which is significantly higher than the incidence of 17% in the general population aged >18 years reported by the National Institute of Mental Health [6,7]. Among these various emotions, we focused on depressed mood, which belongs to depressive episodes. Depressed mood is an emotional state that includes feeling of sadness, irritability, and emptiness.

Up to now, numerous papers exploring the association and risk factors between various types of cancer and depressive episodes have been published. However, the results of these studies remain inconsistent. Several studies have reported significantly higher rates of depression in cancer patients compared to those without cancer [8,9,10,11,12,13]. However, others have identified no association between cancer and depression [14,15,16,17,18]. Most studies have evaluated the association between cancer and depression, a concept encompassing lung, gastrointestinal tract, oral, and breast cancers. In fact, few studies have investigated the relationship of depression with gynecological cancer patients. There are also several studies that have evaluated the association of depression with survival profiles such as disease-free survival (DFS) and overall survival (OS) in cancer patients, but most of these studies have focused on colorectal, lung, and breast cancer, with only one study focusing on cervical cancer [10,19,20,21]. As such, few papers have evaluated the effect of depressed emotional states on survival profiles in gynecological cancers, including cervical cancer, endometrial cancer, and ovarian cancer. Hence, this study aimed to investigate the relationship between gynecological cancer and depressed mood, assess the effects of depressed mood on DFS and OS in gynecological cancer, and identify risk factors, including depressed mood, for cancer aggravation. 

## 2. Materials and Methods

### 2.1. Patients Selection and Data Collection

We retrospectively evaluated the data of patients who had undergone the Patient Health Questionnaire-9 (PHQ-9) depression screening test among patients diagnosed with gynecological cancer at Korea University Anam Hospital from January 2020 to August 2022. 

The PHQ-9 test was used in this study as a self-reported questionnaire to evaluate the severity of depression. It comprises nine items that correspond to the diagnostic criteria for major depressive disorders of the Diagnostic and Statistical Manual of Mental Disorders-5 and determines how often these problems have been experienced in the last two weeks. The nine items of the PHQ-9 depression screening test were as follows: (1). You have little interest or enjoyment in your usual activities; (2). You feel down, depressed, or hopeless; (3). You wake up often or sleep too much because you have difficulty falling asleep or staying asleep; (4). Feeling tired or lacking energy; (5). Lacking appetite or overeating; (6). Hating yourself, seeing yourself as a failure, or thinking you are letting yourself or your family down; (7). Difficulty in concentrating on activities such as reading newspapers and watching television; (8). Movements or speech are too slow, fidgety, or restless, moving around more than usual; and (9). Thinking it would be better if you died, or thinking you would somehow harm yourself. Regarding scoring, 0 indicates the patient never felt an emotion for each item, 1 if the patient felt an emotion for less than a week, 2 if the patient felt an emotion for more than a week, and 3 if the patient felt an emotion for each item almost every day; the total score was then calculated. As a result of the PHQ-9 score, 0–4 points are considered normal or minimal depression, 5–9 points indicate mild depression, 10–14 points indicate moderate depression, 15–19 points are considered moderately severe depression, and ≥20 points indicate severe depression. In this study, 0 points was classified as non-depressed mood (NDM) and ≥1 points were classified as depressed mood (DM) among the score classifications between 0 and 27. 

Age, height, weight, body mass index (BMI), marital status, parity status, menopause status, drinking status, smoking status, occupation status, health insurance status, economic support status, religious status, and number of underlying diseases were identified through anonymized medical information to compare and analyze the characteristics of the participants in the NDM and DM groups. Among the above factors, we compared the social and economic environment characteristics of the study population using the following factors: occupational status, health insurance status, and economic support status. To identify the cancer status and treatment progression status between the two groups, information on the following factors was obtained through medical records: cancer type, cancer stage, radiation therapy (RTx.) status, number of chemotherapy (CTx.) cycles, and status after CTx. Additionally, the timing of the PHQ-9 test was classified as follows to compare and analyze whether the timing of the depression screening test by patients caused a significant difference in the test results: timing of cancer work-up (w/u), timing of cancer operation, timing of CTx. start, and the timing of cancer aggravation. For survival analysis, DFS was calculated as the time from the date of cancer diagnosis to the date of cancer aggravation confirmed by imaging, and OS was defined as the time from the date of cancer diagnosis to the last follow-up period or death. 

### 2.2. Statistical Analysis

A Student’s *t*-test was performed to compare and analyze continuous variables, and the chi-square test or Fisher’s exact test was performed to compare and analyze categorical variables. PFS and OS were analyzed using the Kaplan–Meier method, and the time-to-event outcome was compared with the log-rank test. Cox proportional hazards for multivariate analysis adjusted for factors that were statistically significant as a result of univariate analysis were performed to identify risk factors associated with gynecological cancer aggravation and OS. Statistical significance was defined as a *p*-value < 0.05. All analyses were performed using SPSS statistics for Windows (version 25.0; SPSS Inc., Chicago, IL, USA).

### 2.3. Ethics

The study protocol and waiver of informed consent were approved by the Institutional Review Board (IRB) of Korea University Anam Hospital (IRB number: 2023AN0109). All the procedures were performed in accordance with relevant institutional guidelines and regulations. 

## 3. Results

The overall average age of the patients selected for this study was 57.73 ± 14.42 years, and the average PHQ-9 score was 2.22 ± 4.30 (scores 0–27). Of the 216 patients, 129 (59.4%) belonged to the NDM group (PHQ-9 score = 0), and 88 (40.6%) comprised the DM group (PHQ-9 score ≥1). The comparison of characteristics between the two groups is presented in Table 1. The proportions of patients with ovarian cancer, cervical cancer (*p* = 0.003), stage 2, 3, or 4 (*p* = 0.001), those who had undergone RTx. (*p* = 0.033), those who experienced side effects after CTx. (*p* = 0.001), and those who showed cancer aggravation (*p* = 0.008) were significantly higher in the DM group compared to the NDM group. Conversely, the proportion of patients who underwent cancer w/u (*p* < 0.001) and cancer operation (*p* = 0.036) was significantly higher in the NDM group than in the DM group. There were no statistically significant differences between the NDM and DM groups in terms of occupational status, health insurance status, economic support status, and religious status. Other detailed baseline characteristics and Cox proportional hazards of DFS in univariate analysis for each group not included in Table 1 are presented in the Appendix A.

The mean follow-up period for the study participants was 20.26 ± 12.02 months. Differences in DFS and OS according to the presence or absence of depressed mood were analyzed using Kaplan–Meier curve analysis; the results are presented in Figure 1. DFS was reduced in the DM group compared with the NDM group (*p* = 0.006). The 2-year DFS rate was 93.6% in the NDM group and 86.4% in the DM group. However, the 2-year OS rate was 98.3% in the NDM group and 98.7% in the DM group. There was no statistically significant difference in OS between the two groups (*p* = 0.128). These data suggested that depressed mood was associated with unfavorable DFS in gynecological cancer patients.

To identify possible risk factors of cancer aggravation, univariate analysis and multivariate analysis using Cox proportional hazards was performed (Table 1 and Table 2). Univariate analysis using the Cox proportional hazards model for DFS revealed that menopausal status (hazard ratio (HR), 2.47; 95% confidence intervention (CI), 1.12, 5.44; *p* = 0.025), increased number of CTx. (HR, 1.07; 95% CI, 1.04, 1.09; *p* < 0.001), particularly in cases where CTx. Was administered more than five times (HR, 2.80; 95% CI, 1.59, 4.92; *p* < 0.001), presence of side effects after CTx. (HR, 2.38; 95% CI, 1.28, 4.43; *p* = 0.006), and existence of a depressed mood (HR, 2.05; 95% CI, 1.22, 3.45; *p* = 0.007) were potential risk factors for cancer aggravation. We performed a multivariate Cox regression analysis, adjusted for menopausal status and age; the results are presented in Table 2. The analysis revealed that the risk factors associated with the aggravation of gynecological cancer included stage 3 (HR, 5.43; 95% CI, 2.51, 11.72; *p* < 0.001), stage 4 (HR, 9.11; 95% CI, 4.13, 20.09; *p* < 0.001), administration of CTx. more than five times (HR, 1.07; 95% CI, 1.53, 4.86; *p* = 0.001), the occurrence of side effects after CTx. (HR, 2.07; 95% CI, 1.09, 3.90; *p* = 0.025), and the presence of a depressed mood (HR, 2.09; 95% CI, 1.24, 3.54; *p* = 0.006). On the other hand, cervical cancer (HR, 0.37; 95% CI, 0.19, 0.76; *p* = 0.006) and endometrial cancer (HR, 0.36; 95% CI, 0.18, 0.71; *p* = 0.003) were inversely associated with the risk of cancer aggravation.

## 4. Discussion

In this study, we found that: (I) Depressed mood was associated with unfavorable DFS in gynecological patients. (II) Not only depressed mood, but also stage 3 or 4, having more than five CTx., and experiencing side effects post-CTx. predicted an increased risk of cancer aggravation in gynecological cancer patients.

Emotional disorders like depression in cancer patients can negatively impact outcomes, as this study demonstrates. Although the mechanisms linking cancer and mood disorders are not fully clear, several hypotheses suggest a connection, as represented in Figure 2. Tumor cells secrete cytokines and other compounds, even in tumor cells that have been killed by CTx. or RTx [22,23,24]. These secreted cytokines activate indoleamine 2,3-dioxygenase (IDO), which converts tryptophan, a serotonin precursor, into neurotoxic metabolites like kynurenine (KYN). This process reduces serotonin, increases KYN, which turns into an N-methyl-D-aspartate receptor agonist, spurring glutamate production, oxidative stress, and astrocyte apoptosis. These changes disrupt neurotransmitter metabolism, neural plasticity, and neuroendocrine function, potentially causing depression [22,25,26]. Furthermore, the presence of a depressed mood disorder can lead to an increase in pro-inflammatory cytokines, potentially triggering mechanisms contributing to mood disorders [22]. In other words, it can be said that the molecular environment of cancer and depressed mood disorder create a relationship of bidirectional interactions with each other. Based on the mechanisms introduced earlier, cytokines are secreted from proliferated tumor cells due to advanced cancer stages and cancer aggravation, as well as from tumor cells killed by RTx. or CTx. This can potentially construct a molecular environment conducive to depression. Therefore, our study’s findings that showed a higher incidence of depressed mood in the patients with advanced gynecological cancer at stage 3 or higher, those who have undergone RTx., those who have received CTx. five or more times, and those with cancer aggravation, carry a considerable degree of validity.

Multiple studies have investigated the relationship between cancer and depressive disorder. For instance, Bodurka-Bevers et al. found a 21% depression rate among 246 ovarian cancer patients [15]. Similarly, Britbart et al. reported a 17% depression prevalence and a preference for euthanasia at the same rate among terminally ill cancer patients [11]. Research on head and neck cancer patients showed a 22.2% depression rate [12], and Ciaramella et al. revealed a 49% depression rate in various cancer patients, with a notably higher metastasis rate among the depressed group [13]. These rates exceed the 17% depression prevalence in the general adult population [6,7]. Our study found a 40.6% depressed mood prevalence in gynecological cancer patients, higher than the adult population rate and particularly elevated in stage 2,3, and 4 patients, aligning with Ciaramella et al.’s findings of increased depression in metastatic cases.

On the other hand, some research contrasts with our findings, reporting no significant association between cancer and depression. A study by Lansky et al. on 500 patients with various types of cancer reported a notably lower depression rate of 5.3%, and no correlation with cancer stage or between cancer patients and depression [14]. Similarly, a study by Hong and Tian on 1217 Chinese cancer patients found no significant relationship between cancer progression and depression [15]. A retrospective Italian study of 567 patients with multiple cancers, including breast, rectal, lung, and gynecological cancers, found no significant association between metastatic advanced cancer and depression [16]. In a prospective study involving breast cancer patients, after adjusting for age, the Cox proportional HR analysis for patients with stage 2 or higher breast cancer reported no associated between depression and patient mortality [17]. Additionally, in a nationwide retrospective cohort study conducted in Denmark, the relationship between depressive disorders present before a breast cancer diagnosis and those emerging after the diagnosis on cancer aggravation was analyzed. The findings confirmed that there was no association between depressive disorders and the aggravation of breast cancer [18]. Several factors might explain these differing outcomes. First, unlike our study focusing solely on gynecological cancers, the other studies included a broader array of cancers, potentially affecting the results. Secondly, the different depression screening tools used—PHQ-9 in our study, Hamilton rating scale by Lansky et al., and hospital anxiety and depression scales (HADS) by Hong and Tian—might also contribute to the observed discrepancies.

Several studies have confirmed a significant relationship between depressed state and cancer survival [10,19,20,21]. Sharma et al. found a marked increase in postoperative morbidity in colorectal patients with higher depression scores (*p* = 0.007) [19]. A study by Sullivan et al. revealed increased mortality in lung cancer patients with depression symptoms (HR, 1.17; 95% CI, 1.03, 1.32; *p* = 0.01), particularly those with new-onset (HR, 1.50; 95% CI, 1.12, 2.01; *p* = 0.006) or persistent depression (HR, 1.42; 95% CI, 1.15, 1.75; *p* = 0.001) [20]. Similarly, a study involving breast cancer patients found that both pre-existing depression (HR, 1.35; 95% CI, 1.02, 1.78) and depression developed post-diagnosis elevated patient mortality (HR, 2.42; 95% CI, 1.24, 4.70) [21]. A study with cervical cancer patients showed no significant difference in disease-free survival (DFS) between groups with and without depression (*p* = 0.101), yet overall survival (OS) was notably shorter in the group with depression (*p* = 0.037) [10]. Despite such correlations reported in various cancers, data on gynecological cancer remains limited. Therefore, our study is noteworthy, finding shorter DFS in gynecological cancer patients with depression, though OS showed no significant difference between groups.

This study had several limitations. First, as a retrospective study utilizing medical information, it may not fully represent all gynecological cancer patients. Second, contrary to the diagnostic criteria for depression that classify PHQ-9 scores of 5–9 as mild depression, 10–14 as moderate depression, 15–19 as moderately severe depression, and 20 or above as severe depression, our study differentiated between groups based on the presence of a depressed mood by classifying a PHQ-9 score of 0 as the NDM group, and scores of 1 or above as the DM group. Consequently, our findings may differ from those of studies conducted on patients diagnosed with depression based on the PHQ-9 test results. Third, since the PHQ-9 depression screening test has a questionnaire format, the patients’ responses are based on their personal recollections, which could introduce recall bias. Fourth, the patients who did not fully complete the depression screening questionnaire or refused to participate were excluded from the study, potentially resulting in exclusion bias due to the omitted participants. Nevertheless, the strengths of this study are the association between gynecological cancer and depressed mood, the assessment of the impact of depressed mood on gynecological cancer survival, and the identification of which factors, including depressed mood, contribute to cancer aggravation. 

## 5. Conclusions

The DFS of gynecological cancer patients with depressed mood was significantly shorter than that of the non-depressed group, and the risk of cancer aggravation was significantly increased in the patients with stage 2 or more advanced gynecological cancer, 5 or more multiple CTx., side effects after CTx., and depressed mood. Therefore, appropriate treatment of depressed mood in patients with the above risk factors may contribute to reducing the risk of cancer aggravation in patients with gynecological cancer. Further evaluation is needed to assess whether appropriate psychiatric treatment for gynecological cancer patients with depressed mood, followed by longitudinal follow up, has a positive effect on cancer prognosis.

## Figures and Tables

**Figure 1 ijerph-20-06874-f001:**
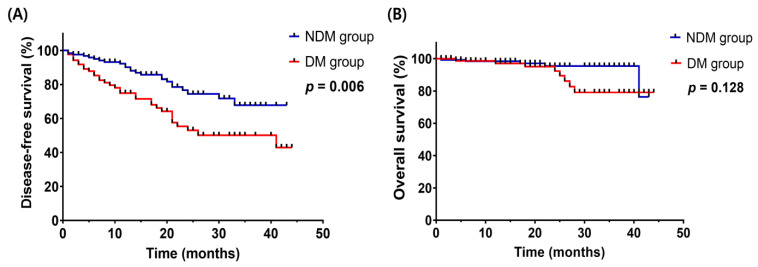
Survival plot by depressed mood status according to the Cox proportional hazard model. (**A**) Disease-free survival plot (**B**) Overall survival plot.

**Figure 2 ijerph-20-06874-f002:**
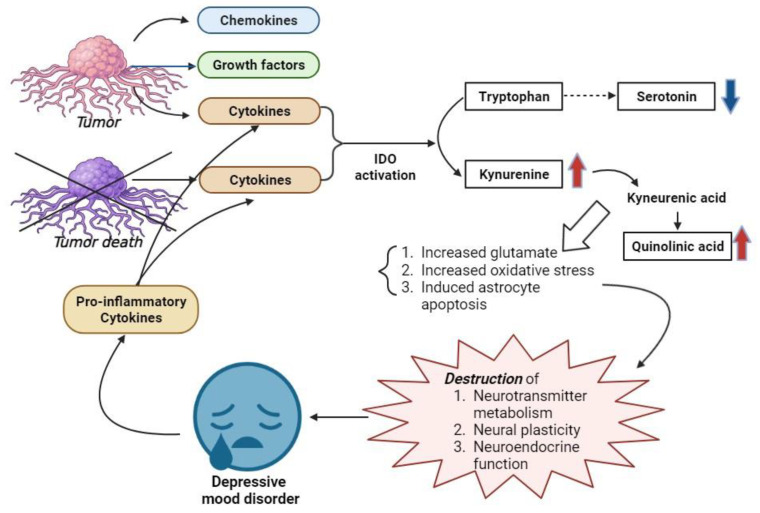
Schematic representation of the mechanism between cancer and depressive mood disorders. Note: IDO, indoleamine 2,3-dioxygenase.

**Table 1 ijerph-20-06874-t001:** Comparison of characteristics between the two groups and Cox proportional hazards of DFS in univariate analysis.

	Non-DepressedMood (N = 129)	Depressed Mood (N = 88)	*p*-Value	Risk Factors for DFS		
				HR	95% CI	*p*-Value
Age (years), M ± SD	57.57 ± 14.23	57.83 ± 14.73	0.898	1.01	(0.99, 1.03)	0.171
Marital status, N (%)			0.635			
Unmarried	19 (14.7%)	16 (18.2%)		1.00	Reference	
Married	100 (77.5%)	66 (75.0%)		1.99	(0.84, 4.69)	0.116
Divorced	5 (3.9%)	1 (1.1%)		2.47	(0.50, 12.32)	0.270
Widowed	5 (3.9%)	5 (5.7%)		1.52	(0.30, 7.58)	0.661
Occupational status, N (%)			0.186			
Inoccupation	30 (23.3%)	21 (23.9%)		1.00	Reference	
Housewife	86 (66.7%)	51 (58.0%)		1.00	(0.55, 1.82)	0.994
Employee	10 (7.8%)	10 (11.4%)		0.44	(0.13, 1.50)	0.188
Profession	3 (2.2%)	2 (2.3%)		2.05	(0.47, 8.99)	0.340
Others	0 (0.0%)	4 (4.4%)		0.00	(0.00, 1.43)	0.972
Health insurance status, N (%)			1.000			
Health insurance	124 (96.1%)	84 (95.5%)		1.00	Reference	
Medical benefit	5 (3.9%)	4 (4.5%)		0.43	(0.06, 3.13)	0.406
Economic support status, N (%)			0.688			
No	126 (97.7%)	85 (96.6%)		1.00	Reference	
Yes	3 (2.3%)	3 (3.4%)		0.68	(0.09, 4.92)	0.703
Religious status, N (%)			0.482			
Atheism	62 (48.0%)	44 (50.0%)		1.00	Reference	
Christianism	33 (25.6%)	18 (20.5%)		0.77	(0.39, 1.54)	0.456
Buddhism	21 (16.3%)	15 (17.0%)		1.01	(0.47, 2.17)	0.987
Catholicism	11 (8.5%)	6 (6.8%)		0.77	(0.26, 2.23)	0.629
Others	2 (1.6%)	5 (5.7%)		3.52	(0.65, 18.99)	0.143
Menopause status, N (%)			0.566			
No	35 (27.3%)	21 (23.9%)		1.00	Reference	
Yes	93 (72.7%)	67 (76.1%)		2.47	(1.12, 5.44)	0.025
Cancer types, N (%)			0.003			
Ovarian cancer	44 (34.1%)	40 (45.5%)		1.00	Reference	
Cervical cancer	36 (27.9%)	34 (38.6%)		0.34	(0.17, 0.68)	0.002
Endometrial cancer	48 (37.2%)	14 (15.9%)		0.32	(0.16, 0.64)	0.001
Uterine sarcoma	1 (0.8%)	0 (0.0%)		0.00	(0.00, 0.00)	0.976
Cancer stage, N (%)			0.001			
Stage 1	72 (55.8%)	27 (30.7%)		1.00	Reference	
Stage 2	20 (15.5%)	17 (19.3%)		0.03	(0.00, 0.26)	0.001
Stage 3	25 (19.4%)	22 (25.0%)		0.05	(0.01, 0.42)	0.006
Stage 4	12 (9.3%)	22 (25.0%)		0.17	(0.02, 1.34)	0.093
RTx. status, N (%)			0.033			
No	124 (96.1%)	78 (88.6%)		1.00	Reference	
Yes	5 (3.9%)	10 (11.4%)		0.77	(0.24, 2,48)	0.666
Number of CTx.	5.96 ± 8.20	3.97 ± 5.44	0.032	1.07	(1.04, 1.09)	<0.001
Categorization of number of CTx.			0.047			
CTx. < 5, N (%)	63 (48.8%)	55 (62.5%)		1.00	Reference	
CTx. ≥ 5, N (%)	66 (51.2%)	33 (37.5%)		2.80	(1.59, 4.92)	<0.001
Side effect after CTx., N (%)			0.001			
No	122 (94.6%)	71 (80.7%)		1.00	Reference	
Yes	7 (5.4%)	17 (19.3%)		2.38	(1.28, 4.43)	0.006
Timing of cancer w/u, N (%)			<0.001			
No	42 (32.6%)	60 (68.2%)		1.00	Reference	
Yes	87 (67.4%)	28 (31.8%)		0.60	(0.35, 1.01)	0.053
Timing of cancer operation, N (%)			0.036			
No	100 (77.5%)	78 (88.6%)		1.00	Reference	
Yes	29 (22.5%)	10 (11.4%)		0.55	(0.25, 1.20)	0.132
Timing of CTx. start, N (%)			0.197			
No	123 (95.3%)	80 (90.9%)		1.00	Reference	
Yes	6 (4.7%)	8 (9.1%)		0.96	(0.30, 3.10)	0.951
Cancer aggravation, N (%)			0.008			
No	103 (79.8%)	56 (63.6%)				
Yes	26 (20.2%)	32 (36.4%)				
Death, N (%)			0.105			
No	124 (96.1%)	80 (90.7%)				
Yes	5 (3.9%)	8 (9.3%)				
Depressed mood						
No				1.00	Reference	
Yes				2.05	(1.22, 3.45)	0.007

Note: Values are presented as mean ± standard deviation or N (%). M ± SD, mean ± standard deviation; DFS, disease-free survival; HR, hazard ratio; CI, confidence interval; RTx., radiation therapy; CTx., chemotherapy; w/u, work up.

**Table 2 ijerph-20-06874-t002:** Cox proportional hazards of disease-free survival in multivariate analysis.

	Risk Factors for DFS		
	HR *	95% CI	*p*-Value
Cancer type			
Ovarian cancer	1.00	Reference	
Cervical cancer	0.37	(0.19, 0.76)	0.006
Endometrial cancer	0.36	(0.18, 0.71)	0.003
Uterine sarcoma	0.00	(0.00, 0.00)	0.977
Cancer stage			
Stage 1	1.00	Reference	
Stage 2	1.50	(0.54, 4.13)	0.436
Stage 3	5.43	(2.51, 11.72)	<0.001
Stage 4	9.11	(4.13, 20.09)	<0.001
Number of CTx.	1.07	(1.04, 1.09)	<0.001
CTx. < 5	1.00	Reference	
CTx. ≥ 5	2.72	(1.53, 4.86)	0.001
Side effect= after CTx.			
No	1.00	Reference	
Yes	2.07	(1.09, 3.90)	0.025
Timing of cancer w/u			
No	1.00	Reference	
Yes	0.63	(0.37, 1.08)	0.092
Timing of cancer operation			
No	1.00	Reference	
Yes	0.59	(0.27, 1.30)	0.188
Timing of CTx. start			
No	1.00	Reference	
Yes	0.81	(0.25, 2.63)	0.729
Depressed mood			
No	1.00	Reference	
Yes	2.09	(1.24, 3.54)	0.006

Note: DFS, disease-free survival; OS, overall survival; HR *, hazard ratio adjusted for age and menopausal status; CI, confidence interval; BMI, body mass index; RTx., radiation therapy; CTx., chemotherapy; w/u, work up.

## Data Availability

Data sharing is not applicable to this study because of privacy or ethical restrictions. The data will be shared on reasonable request to the corresponding author.

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
