# Peer review of "Depressed Mood as a Significant Risk Factor for Gynecological Cancer Aggravation"

_ijerph, 2023, doi:10.3390/ijerph20196874_

Round 1
Reviewer 1 Report
The authors address an absolutely relevant issue in the repercussion of emotional life in the course of gynecological oncological processes.
We believe that the authors should analyze the characteristics of their population in regards to their social, economic, cultural and religious environment as conditions of their emotional situation and coping with the evolutionary process of the gynecological oncological process. Circumstances such as experiencing and coping with a poor prognosis or advanced stage or progression of the disease must be treated in detail (these conditions accumulate a greater number of patients with emotional disorders).
We recommend that the authors clarify their statements about the direct impact of depressive symptoms and poor prognosis. They should separate between the population of initial and advanced stages in an attempt to delimit the authentic repercussion of emotional life and the evolution of the tumor process.
Author Response
Response to Reviewer 1 Comments
Dear reviewers
Thank you for giving us the opportunity to submit a revised draft of the manuscript “ Depressed Mood as a Significant Risk Factor for Gynecological Cancer ” for publication in the ijerph. We appreciate the time and effort that you and the reviewers dedicated to providing feedback on our manuscript and are grateful for the insightful comments on and valuable improvements to our paper.
We have incorporated most of the suggestions made by the reviewers. Any revisions to the manuscript be marked up using the track changes function at MS Word. In addition, the changed text color was changed to blue and displayed. Please see below, for a point-by-point response to the reviewers’ comments and concerns.
Point 1: We believe that the authors should analyze the characteristics of their population in regards to their social, economic, cultural and religious environment as conditions of their emotional situation and coping with the evolutionary process of the gynecological oncological process. Circumstances such as experiencing and coping with a poor prognosis or advanced stage or progression of the disease must be treated in detail (these conditions accumulate a greater number of patients with emotional disorders).
Response 1: Thanks for the good advice. As you mentioned, the social, economic, and religious environments of the subbjects in this study are factors that may affect their depressed mood, so I agree that it is necessary to compare theses factors in the characteristics. In order to compare the social and economic environment of our subjects, we compared their occupational status, health insurance status, and economic support status, and the results are shown in Table S1 in the supplementary materials. As you suggested, to compare the religious environment, I added the religious status of each subject to compare the religious status between Non-depressed mood and Depressed mood groups. The comparative results for each group for these social, economic, and religious environments have been modified by adding them to Table 1. In the Materials and Methods section, we also describe the use of factors such as occupational status, health insurance status, and economic support to characterize the socioeconomic environment of the study population.
Point 2: We recommend that the authors clarify their statements about the direct impact of depressive symptoms and poor prognosis. They should separate between the population of initial and advanced stages in an attempt to delimit the authentic repercussion of emotional life and the evolution of the tumor process.
Response 2: In this study, the stage of gynecological cancer patients was divided into stage 1, 2, 3,and 4, and the proportion distribution between NDM group and DM group was compared. In general, the early stage of gynecological cancer includes stage 1 and stage 2, and the advanced stage includes stage 3 and stage 4. In other words, the analysis results of this study, which expressed the proprtion of gyencological cnacer as stage 1, 2, 3, and 4, included the classification of early stage and advanced stage. Therefore, it is thought that this part of the results in Table 1 represents the difference in depressed mood between early and advanced stages of the cancer you mentioned.
Reviewer 2 Report
The Authors demonstrate results in which they tried to answer important questin: is there a relationship between depressed mood and gynecological cancer outcomes. They performed quite advanced statistical analysis using two cohorts: non-depressed mood and depressed mood. The results they obtained are interesting and can be published. Below my remarks:
- the literature survey is quite short, only 23 positions. At the same time they write in line 61-62 that numerous papers exploring the association and risk factors between various types of cancer and depressive episodes have been published." I understand that the reoults remain inconsistent, however the literature review could be richer
- Figure 1, please enlarge it and incerase quality
- The "Discussion" is very good, however the "Conclusions" is very short. Please elaborate more - what can be done after this reasearch? can any experiments can be done?
The language is fine.
Author Response
Response to Reviewer 2 Comments
Dear reviewers
Thank you for giving us the opportunity to submit a revised draft of the manuscript “ Depressed Mood as a Significant Risk Factor for Gynecological Cancer ” for publication in the ijerph. We appreciate the time and effort that you and the reviewers dedicated to providing feedback on our manuscript and are grateful for the insightful comments on and valuable improvements to our paper.
We have incorporated most of the suggestions made by the reviewers. Any revisions to the manuscript be marked up using the track changes function at MS Word. In addition, the changed text color was changed to blue and displayed. Please see below, for a point-by-point response to the reviewers’ comments and concerns.
Point 1: The literature survey is quite short, only 23 positions. At the same time they write in line 61-62 that numerous papers exploring the association and risk factors between various types of cancer and depressive episodes have been published." I understand that the results remain inconsistent, however the literature review could be richer.
Response 1:Thanks for the great feedback. As you mentioned, there are several papers that have found an association between cancer and depressive episodes, and I agree that the references in the manuscript are a bit insufficient. Therefore, I have revised the manuscript to ad references that states that there is no association between cancer and depresive episodes. For a detailed explanation of the papers that found no association between cancer and depression, I revised them as described in the discussion part.
Point 2: Figure 1, please enlarge it and incerase quality.
Response 2: Thanks for the good advice. As per your advice, I have increased the resolution of figure 1 and ennlarged the size of figure 1.
Point 3: The "Discussion" is very good, however the "Conclusions" is very short. Please elaborate more - what can be done after this research? can any experiments can be done?
Response 3: Thanks for the good advice. As your suggestion, I have revised the conclusion part to make it clear what it is trying to convey and also included a reference to future research directions.